# Perceptual and Metabolic Responses During Resistance Training Sessions: Comparing Low-Load Plus Blood Flow Restriction with High-Load Plans

**DOI:** 10.3390/sports13050148

**Published:** 2025-05-16

**Authors:** Anderson Geremias Macedo, Danilo Alexandre Massini, Tiago André Freire Almeida, Adriana Teresa Silva Santos, Giovane Galdino, David Michel de Oliveira, Dalton Muller Pessôa Filho

**Affiliations:** 1School of Sciences (FC), São Paulo State University (UNESP), Bauru 17033-360, São Paulo, Brazil; andersongmacedo@yahoo.com.br (A.G.M.); dmassini@hotmail.com (D.A.M.); tiagofalmeida.w@gmail.com (T.A.F.A.); 2Graduate Program in Human Development and Technology, São Paulo State University (UNESP), Rio Claro 13506-900, São Paulo, Brazil; 3Institute of Motricity Sciences, Federal University of Alfenas (UNIFAL), Alfenas 37133-840, Minas Gerais, Brazil; adriana.santos@unifal-mg.edu.br (A.T.S.S.); giovane.souza@unifal-mg.edu.br (G.G.); 4Pos-Graduation Program in Rehabilitation Sciences, Institute of Motricity Sciences, Federal University of Alfenas (UNIFAL), Santa Clara Campus, Alfenas 37133-840, Minas Gerais, Brazil; 5Department of Physical Education, Federal University Jataí (UFJ), Jataí 75801-615, Goiás, Brazil; profdoliveira@ufj.edu.br; 6Postgraduate Program in Animal Bioscience (PPGBA), Institute of Health Sciences (ICS), Federal University of Jataí (UFJ), Jataí 75801-615, Goiás, Brazil

**Keywords:** lactic acid, perceived exertion, blood flow restriction exercise, strength training, male young adults

## Abstract

This study analysed perceived exertion (RPE) and blood lactate ([La^−^]) responses to two resistance training protocols planned with high- (HLI) and low-load intensities combined with blood flow restriction (LLI+BFR). Fourteen trained adults (26.2 ± 2.6 years) performed the HLI and LLI+BFR protocols 48 h apart. The HLI was planned with 70% 1RM (one repetition to maximum), three sets, 12 repetitions, 60 seconds (s) of rest between sets and 120 s between exercises; LLI+BFR was performed at 30% 1RM, for three sets, 15 repetitions, and with 30 s of rest between sets and 180 s between exercises. Blood samples (for [La^−^] analysis) and RPE (Borg 0–10 scale) were collected in the first minute after each exercise. A two-way ANOVA compared RPE and [La^−^] responses between exercises in the same protocol, and between protocols comparing the same exercise. RPE increased from the first to the last half (involving upper-limbs and lower-limbs) of exercises in both protocols (*p* < 0.001). All exercises in HLI elicited higher RPE values than LLI+BFR (*p* < 0.001). Average RPE scores were higher in HLI than for LLI+BFR (8.1 ± 0.6 > 6.2 ± 1.1, *p* < 0.001). The [La^−^] also increased throughout the exercises, with a higher peak response in LLI+BFR than for HLI (9.8 ± 1.6 > 7.2 ± 1.3 mmol × L^−1^, *p* < 0.01). Perceptual and metabolic responses during HLI and LLI+BFR training were distinguishable, despite both protocols characterising a high-intensity stimulus.

## 1. Introduction

Resistance training (RT) planned with high-load intensity (i.e., HLI ≥ 70% of one repetition to maximum, 1RM) and exercises targeting different muscle groups improves muscle control and function [1], strength, and mass [2,3], which are important elements of health as well as motor performance [2]. Similarly, RT planned with low-load intensity (i.e., LLI = 30 ≥ 50% 1RM) provides a comparable effect on muscle anabolic signalling to HLI [4], despite of specific adjustment of neural control and strength (i.e., enhanced muscular endurance) [1,3]. Therefore, training with LLI is highly recommended for individuals at temporary or permanent risk to lifting heavy weights (e.g., low conditioning level, injury rehabilitation, physical impairment, reduced mobility, and metabolic and cardiovascular diseases) [3]. Moreover, RT with LLI combined with blood flow restriction (LLI+BFR) procedures (i.e., 20 to 40% 1RM, 2 to 4 sets, 15 to 30 repetitions per set with rest intervals of 30 to 60 seconds (s), and with continuous or intermittent blood flow restriction at 40 to 80% of arterial occlusion pressure, AOP) [5,6] is considered as effective as HLI training to alter both muscle fibre growth [7] and force [1], and is also recommended as an additional alternative of RT for populations with limited ability to training with heavier loads [8]. Since HLI and LLI+BFR are both considered high-intensity and short-term exercises due to the extensive reliance on type II fibre recruitment and glycolytic activation [7], studies have used internal load variables for the control and comparison of the levels of intensity (e.g., low, moderate, and high) and exertion (e.g., easy, heavy, and hard) of different combinations of training variables [9,10].

The rating of perceived exertion (RPE) and blood lactate concentration ([La^−^]) have been observed to be effective parameters in grading (or controlling) the levels of intensity and exertion during RT [11,12]. Studies involving trained young adults of both sexes have shown that the intensity of RT (i.e., load in percentage of one-repetition to maximum, %1RM) influences RPE and [La^−^], regardless of whether one or more exercises are performed [13]. Indeed, the RPE and [La^−^] responses during a single-exercise (i.e., bicep-curl) performance at 90% 1RM reached higher values than the same exercise performed at 30% 1RM for both RPE and [La^−^] variables (e.g., 13.2 vs. 9.7, and 3.7 vs. 2.5 mmol × L^−1^) [14]. Similarly, higher values of RPE and [La^−^] were reported at 70% 1RM when compared to 50% 1RM (8.8 vs. 5.5, and 8.7 vs. 5.5 mmol × L^−1^) during a training session involving full-body weight-lifting movements [12]. Therefore, load intensity during RT has been reported as the main determinant of RPE and [La^−^] responses during single- and multi-joint RT exercises [5,13].

The assumption underlying the increase in both RPE and [La^−^] values with the increase in load-intensity during RT relies on the stimulation of the central efferent drive at high-load intensity [1,14], which in turn increases the recruitment of muscle fibres at a high firing rate, leading to metabolite accumulation, and thus enhancing the perception of conscious exertion level [9,14]. In contrast, RT with LLI and combined with blood flow restriction (BFR) also increases RPE and [La^−^] responses since occlusion is supposed to enhance the recruitment of type II muscle fibres and reduce oxygen supply to the muscle, with both increasing the demand upon anaerobic glycolysis [15]. In addition, the net increase in [La^−^] stimulates afferent receptors in skeletal muscles [5,15], enhancing sensation of discomfort and pain, and the conscious perception of exertion, which likely increases RPE responses [9,10].

Despite RT with HLI being associated with higher values of RPE and [La^−^] when compared to training with LLI, the training techniques combining LLI and BFR may also elicit elevated RPE and [La^−^] responses, similar to those observed with HLI training [10]. However, the assumptions of similar perceptual and metabolic responses between HLI and LLI+BFR have been supported by protocols involving one to three sets of a single exercise, thus leaving it unclear whether such as pattern of response can also be observed during an entire resistance training session. For example, studies have reported similar RPE (7.9 vs. 6.4) and [La^−^] (4.2 vs. 4.1 mmol × L^−1^) responses when comparing three sets of unilateral knee extension exercises at 80 vs. 20% 1RM (one repetition to maximum) with BFR [15], as well as no significant differences in RPE values (15 vs. 13 and 17) during sets of knee extension exercises performed at different load intensities (e.g., 70, 20, and 30% 1RM) [10].

Indeed, RPE and [La^−^] have shown a consistent tendency to be not different (with few controversies) when comparing perceptual and physiological responses from protocols planned with different loads (e.g., HLI vs. LLI+BFR) and a reduced number of sets [9,10,16], but when performing multiple sets, the RPE and [La^−^] responses have shown a tendency to increase across different loads and pressures [9,10]. In addition, it is still unclear whether this tendency remains irrefutable for an entire training session involving RE for both upper- and lower-limbs (i.e., what about the responses from a training session planning with multiple sets for different muscle groups?). Studies have shown that both RPE and [La^−^] increase progressively throughout the resistance training session due to the involvement of different muscle groups, volume load [11,17], and an accumulation of metabolites [18], suggesting that the responses of RPE and [La^−^] to the training session would be different from those observed during the execution of sets for only one exercise planned with HLI or LLI+BFR.

Actually, the cumulative effects of RPE and [La^−^] over an entire training session with HLI vs. LLI+BFR have been not addressed, which is the gap to be overcome in the current study. Thus, following the hypothetical supposition that each protocol demands a particular level of exertion, when analysed from perceptual and metabolic responses during training sessions involving muscle groups of different body regions, the current study aimed to address RPE and [La^−^] responses to training sessions designed with HLI and LLI+BFR to verify whether previous statements regarding the non-different stimuli remain irrefutable, or whether differences would be discernible. Therefore, this would aim to confirm if the interchangeable usage of both protocols for similar RE training proposes should be recommended. In addition, we expected to verify which training plan variables contribute to perceiving and responding to training as a high-intensity exercise.

## 2. Materials and Methods

### 2.1. Participants

Fourteen recreationally trained men, training for at least six months and three times a week (age: 26.2 ± 2.6 years, height: 177.8 ± 6.7 cm, body weight: 82.6 ± 16.2 kg), were informed about the procedures and signed a form of consent to participate in this study. The auto-declared training routine of the participants at the moment of the study included 3–5 sessions per week, 6–8 exercises per session, 3–4 sets per exercise, and 10–12 repetitions to the maximum level of exertion. Exclusion criteria included diseases such as ischemia, diabetes, arrhythmias, hypertension, and obesity (BMI 30 kg/m^2^) due to the risk of an adverse effect of BFR procedures, as well as due to the limitations of those individuals to exercise with HLI [3,5]. All experiments reported in this manuscript were performed in accordance with the ethical standards of the Helsinki Declaration. This research was approved by the Ethics Committee of Human Research from the São Paulo State University (UNESP), protocol CAEE 19824719.3.0000.539, number 3.572.379.

### 2.2. Strength Measurements and Training Protocols

The 1RM test was applied to assess the reference of maximal load in each exercise planned for both experimental training protocols. Test procedures followed previous recommendations [19], which (briefly) consist of performing only two tests per day (one exercise for the upper- and another for the lower-limb exercise), observing 24 h between each testing session. During the test, the initial load was set as the average scores for strength in upper- and lower-limbs (according to the age, sex, and body mass of the participant). If participants performed two or more repetitions in the first attempt with the initial load for a given exercise, then another attempt was performed with the load progressing by 10%. In turn, if participants were not able to lift the initial load once, then the load was reduced by a similar percentage for the next attempt. Thus, this process was repeated one or two more times up to the maximum weight able to be lifted once (i.e., the 1RM load). Between each attempt a resting interval of 180 s was applied.

Two different protocols (Figure 1) planned with eight exercises were designed for both upper- and lower-limbs to be performed in a randomised order and on different days (48 h apart, see Table 1). The exercises were performed following the same order in both protocols: (Ex.1) flat bench press, (Ex.2) seated row, (Ex.3) triceps curl, (Ex.4) biceps curl, (Ex.5) single leg extension, (Ex.6) prone leg curl machine, (Ex.7) leg press 45°, and (Ex.8) rack calf raises. The protocols were planned following guidelines proposed for muscle mass and strength improvements involving intermediate individual training with HLI [20] and LLI [3,4], which recommend a session with HLI being as follows: load 70 ≥ 85% of 1RM, eight to twelve repetitions per set, two to four sets with one to two minutes of rest between sets, two to four minutes of rest between exercises consisting of multi-joint and single-joint exercises targeting different muscle groups and body segments [20]. For training with LLI, the following procedure was followed: load 20 ≥ 40% of 1RM, 15 to 30 repetitions per set, 2 to 4 sets, rest intervals of 30 to 60 s between sets, and one to two minutes of rest between exercises targeting major and minor muscle groups for upper- or lower-limbs [3,4,20]. For LLI resistance training, with BFR aiming to stimulate muscles to work similarly with high-intensity physiological and perceptual demand, continuous or intermittent blood flow restriction (ranging from 40% to 80% blood flow restriction) has been recommended [6,13,21].

Before the execution of each protocol, the participants received only basic information about the number of repetitions in each set, resting time between sets, and technique and velocity of the exercise performance. During the execution of the protocols, the researchers controlled the number of repetitions, duration of rest, number of sets, and the technique and velocity of movement (i.e., keeping the recommended moderate rate 1:2 s, respectively, for concentric and excentric phases during strength training [2]). All protocols were preceded by a 10 min warm-up (e.g., stretching and joint mobility for upper- and lower-limbs), which excluded exercises able to stimulate blood lactate increase (e.g., cycling, running, and calisthenics).

The BFR procedures were applied by two specific pneumatic cuffs for upper-limbs (80 cm long, 7 cm wide; Scientific-arm-WCS, Cardiomed^®^, Curitiba, Brazil) and for lower-limbs (84 cm long, 12.5 cm wide; Scientific-leg-WCS, Cardiomed^®^, Curitiba, Brazil). According to the limb engaged in the exercise, the cuff was placed bilaterally around the proximal part of the arms and legs. During exercise and rest between sets, the cuff pressure was maintained at 50% of the resting systolic blood pressure and fully deflated during the rest between exercises. Immediately before each exercise, the cuff was inflated, the allocation around the body segment was checked, and participants were questioned about any discomfort [5]. While exercising and resting between sets, cuff pressure, position, and discomfort were all continually checked [6], with the cuff being fully deflated after the third set accomplishment [5] to avoid passive influence of BFR on muscle ischemia and metabolite production while resting between exercises [1]. The choice of using systolic blood pressure (SBP) to determine the cuff pressure during the protocol of training considered certain factors, listed as follows: (i) Doppler measurements show error increasing with user experience and training, equipment technology, and subject vessel morphology [22,23], which might preclude the usability for the training routine in gyms; and (ii) the body position is different while performing each exercise of the protocol, which modifies the arterial occlusion pressure (AOP) accordingly [24], and which might reduce the feasibility of the AOP measurements when training with multiple exercises. Thus, searching for feasibility and considering the assumptions above, SBP was defined as the control variable of exercise intensity also taking into account the association with AOP values [10].

The volume load of each protocol was assessed by multiplying the number of repetitions and load in all the exercises performed in the session, represented in kilograms (kg). The total number of repetitions was calculated by the sum of all repetitions performed during each protocol [12].

### 2.3. Metabolic and Perceptual Measurements

The [La^−^] (in mmol × L^−1^) was analysed from blood samples (25 μL of capillary blood) of the earlobe, using heparinized and calibrated capillaries. The samples were stored in Eppendorf tubes with 50 μL of sodium fluoride (NaF at 1%), and analysed by an enzymatic method (Yellow Spring 2500-STAT). The samples were taken at rest (i.e., before starting the first exercise), immediately after the three sets of each exercise, and at 3, 5, and 7 min during the recovery period of each protocol. The value after three sets of each exercise gave [La^−^] throughout the protocols, and the peak values measured during the recovery gave the [La^−^] peak for each protocol. The RPE scores were obtained with the Borg 0 to 10 scale, from which the participants indicated the number corresponding to the exertion level of each exercise just performed (after three sets, and with the cuff still inflated), giving the RPE scores after each exercise. The average RPE for the entire training protocol was obtained from the average of the scores verbally indicated by the participants after each exercise [12,14], i.e., the RPE protocol (RPE-P).

### 2.4. Statistical Analysis

Data are shown by mean ± SD, and normality and sphericity were verified with the Shapiro–Wilk and Mauchly tests, respectively. In the absence of data sphericity, the Greenhouse–Geisser correction was adopted. The independent (two-tailed) Student *t*-test verified the differences between the training protocols (volume load, number of repetitions, [La^−^] peak, and RPE-P scores). Two-way analysis of variance (ANOVA—exercises and protocols, with Bonferroni as post hoc) was applied to assess differences in the responses of RPE and [La^−^] between exercises conditions, while also accounting for repeated measures of these variables, ensuring appropriate statistical control. The variability in the average responses for each protocol was used a 95% confidence interval (95% CI). The effect size (ES) for the Student *t*-test was determined by Hedges’ *g*, and the values were interpreted as follows: <0.19 [trivial], 0.20–0.49 [small], 0.50–0.79 [medium], 0.80–1.29 [large], and >1.30 [very large] [25]. For ANOVA, the ES was the partial eta square (η^2^*p*), which was interpreted as follows: 0.0099 [small], 0.0588 [medium], and 0.1379 [large] [25]. The sample power (SP) observed for the comparisons was determined by G*Power 3.1 software, using a 95% confidence level (Z_1_-α/2 = 1.960), and a satisfactory power of 85% (Z_1_-β = 1.036) for all analyses with N = 14. This sample size was also estimated using G*Power 3.1 software with the following input parameters: significance level at *p* = 0.05 (Z_1_-α/2 = 1.960) with 90% power (Z_1_-β = 1.282), and a large effect size to be detected (Hedges’ *g* ≥ 1.3) for the comparisons of RPE and [La^−^] average values between HLI and LLI+BFR protocols. Correlations between the responses of RPE and [La^−^] to the performance of the exercises in the same protocol, and between the performance of each pair of exercises in different protocols, as well as the correlation of both these variables with training variables (load, number of repetitions, and protocol duration), were analysed by Pearson’s coefficient. The significance level adopted for all analyses was α < 0.05.

## 3. Results

Figure 2 shows the volume load (Panel A) and total number of repetitions (Panel B) for each RT protocol. The comparison showed that volume load is higher for HLI than for LLI+BFR (t_[26]_ = 6.385, *p* < 0.001, *g* = 2.34 [ES = very large], SP = 99%), whereas the number of repetitions was significantly lower for HLI than for LLI+BFR (t_[16.5]_ = −19.292, *p* < 0.001, *g* = 7.08 [ES = very large], SP = 100%). Hence, the HLI protocol demanded a higher training load and shorter performance than LLI+BFR. Figure 3 shows the values of RPE-P and [La^−^] peak responses after each protocol performance. The average values of RPE-P (Panel A) were significantly higher in HLI compared to LLI-BFR (t_[26]_ = 5.621, *p* < 0.001, *g* = 2.03 [ES = very large], SP = 99%), and [La^−^] peak response was significantly higher in LLI+BFR compared to HLI (Panel B, t_[26]_ = −4.740, *p* < 0.001, *g* = 1.74 [ES = very large], SP = 99%). Therefore, HLI protocol performance was perceived as harder despite being metabolically less heavy than LLI+BFR.

Table 2 compares the values of RPE and [La^−^] responses between the exercise sequence in each protocol. The RPE differed between exercises of both protocols (F_[7, 182]_ = 14.767, *p* < 0.001, η^2^*p* = 0.362 [ES = large], SP = 99%), suggesting that both protocols were performed with high and increased scores of RPE. A similar tendency was observed for [La^−^] values (F_[8, 208]_ = 148.6, *p* < 0.001, η^2^*p* = 0.851 [ES = large], SP = 100%), suggesting that metabolic demand increased throughout exercise performance for both protocols. However, these tendencies of RPE and [La^−^] responses to increase across the exercises sequence (from E1 to E8) in each protocol were significant only between [La^−^] responses to the exercise sequence from E1 to E6, but not between exercises E6 to E8 (in which large muscle groups in the lower-limbs are involved) (Table 2). In turn, the RPE responses differed when comparing the score for exercise E1 with the scores for exercises E4 to E7, which is also a sequence of exercises mostly involving large muscle groups in the lower-limbs (Table 2).

The comparison of the exercises between each training condition is shown in Figure 4. The results demonstrate that each exercise differed from the respective pair between protocols regarding the responses of RPE (F_[1, 26]_ = 30.963, *p* < 0.001, η^2^*p* = 0.544 [ES = large], SP = 100%) and [La^−^] (F_[1, 26]_ = 16.016, *p* < 0.001, η^2^*p* = 0.381 [ES = large], SP = 97%), suggesting that protocols differed completely regarding RPE and [La^−^] responses whatever the exercise performed. Indeed, all RPE scores from E1 to E8 in HLI were higher than those for LLI+BR, and [La^−^] responses showed an inverse trend, i.e., all responses from E1 to E8 in HLI were lower than those for LLI+BFR (Table 1, *p* < 0.05 for all comparisons).

Correlations were observed between the responses of RPE-P and [La^−^] during the execution of each protocol, which range from r = −0.46 to −0.69 (*p* < 0.01) when eliminating the influence of the volume-load or the time duration of the protocols, respectively. Thus, the perceptual level of effort is inversely associated with the level of metabolic activation. The responses of RPE and [La^−^] during the execution of each pair of exercises in different protocols were also correlated (r = 0.53 and 0.75, *p* < 0.05), with the RPE-P responses in either HLI or LLI+BFR protocols correlated to the number of repetitions in each protocol (r = −0.52 and −0.61, *p* < 0.05). Therefore, there is a tendency to perceive a given exercise as harder or to demand high metabolic activation whatever the protocol, with the training performed for a shorter length of time being harder. The remaining correlations tested did not attain a significant level (i.e., RPE vs. other training variables except for repetition number, and [La^−^] vs. training variables).

## 4. Discussion

The current findings evidenced that RPE and [La^−^] responses were not aligned in graduating the level of exertion of the HLI and LLI+BFR protocols, indicating that training with HLI was perceived as heavier than LLI+BFR, while the metabolic response from [La^−^] measurement indicated the LLI+BFR training as the harder protocol instead of HLI. Therefore, these findings are not aligned with previous studies reporting no significant differences in RPE and [La^−^] during RT planned with a unique exercise, different intensities of load, and BFR pressures. Moreover, the responses of RPE and [La^−^] were observed to be negatively related when performing the exercises of each protocol, suggesting that the training condition with high tensional demand did not elicit high metabolic stress.

Previous studies have indeed reported controversial RPE responses during RT planned with a unique exercise (e.g., knee-extension) and performed with different load intensities. For example, higher RPE scores (7.9 vs. 6.4) have been reported for LLI+BFR than HLI (at 20% 1RM + 100% AOP vs. 80% 1RM) [15], while no different RPE scores (11 vs. 13 vs. 12) were reported when comparing LLI+BRF to HLI (at 20 or 30% 1RM + 40% AOP vs. 70% 1RM) [10]. Indeed, the differences in RPE responses (if observed) while training with different load intensities and BFR procedures have no tendency to be large, despite increases in RPE score being associated with volume (repetitions and sets), load intensity (heavy weight), and cuff pressure (high percentage of partial blood occlusion) [10,26]. This tendency was strengthened by the current findings, which suggested that training conditions planned with high-load intensity were a distinguishable effort from a low-load intensity (given the differences in the RPE scores), and also reinforces the assumption that the heaviest load demands fewer repetitions is perceived as harder, and stimulates muscle adjustments to improve the performance specifically in high-intensity tasks [1,2,3].

In fact, there are different psychological factors influencing the heterogeneity of RPE during RT, and hence the assumption that training with resistance exercise is perceived as harder as load and volume increase have shown conflicting results [27,28]. For example, individuals experiencing long-term training with a heavy load have a tendency to perceive low load-intensity training with BFR as more difficult than high load-intensity training (e.g., 50% 1RM + 110 mmHg vs. 80% 1RM with RPE = 9 vs. 6, respectively), in spite of the higher-volume load of the high-load intensity training (197 vs. 300 kg, respectively) [29]. It is probable that the discomfort faced by an unusual (or novelty) training condition (e.g., low load performed to failure) together with the cuff pressure might be a factor contributing to the increase in RPE response, as demonstrated for leg exercise (e.g., knee extension and flexion) performed at 20% (+BFR at 80% AOP) vs. 80% 1RM, in which the RPE score was higher for low- than for high-load intensity [30]. However, for exercise involving arms (e.g., arm curl and triceps extension), this same study did not evidence a difference in RPE responses.

Thus, the body region involved in exercise also influences RPE scores, with studies supporting the notion that flow restriction in body regions with larger muscles finds ischemia harder than for areas with small muscles [31,32]. This assumption is, however, partially aligned with the current results, since the observed RPE scores showed a trend to increase between the set of exercises performed with upper- and lower-limbs in both protocols. Moreover, when comparing the RPE responses between the current protocols, the same exercises planned for arms and legs is perceived as harder to be performed for the HLI protocol than LLI+BFR. Therefore, the current study contributed to the assumption that resistance exercises are perceived harder when performed with large muscle groups and with a high-intensity load, which was especially observed for a complete training section (i.e., multiple sets for exercises involving small and large muscles).

However, the higher RPE response in the current study for the HLI protocol might also be a result influenced by the possible occurrence of failure while performing the exercises of this protocol, despite both protocols not being planned to be performed until volitional fatigue, since this condition trends to drive perceptual responses to maximal scores, thus limiting the observation of differences between protocols [33]. Indeed, for protocols ensuring repetition-to-failure, higher RPE values have been reported for a high- than low (+BFR)-load protocol [10,26]. Anyway, the role of the repetition-to-failure mode of training on RPE response is conflicting. For example, while some studies have reported higher RPE responses to low- (+BFR) than to high-load intensity when exercise in each protocol is performed to failure [23,26], there are other studies reporting the lack of differences on RPE responses [27,34], or rather higher RPE scores for high- than low (+BFR)-load intensity protocols [35,36].

Interestingly, for the studies showing that RPE responses did not differ between protocols with different load intensities, or which favour exercise performed with high-load intensity, there is evidence that the volume load of exercise is often higher for the high-load intensity protocol (e.g., for a young group, the RPE = ~18 for both LLI+BFR and HLI protocols, where volume load = 2519 kg vs. 4283 kg, respectively [34]). This tendency is further supported by the assumption that either the repetition-to-failure mode or a high-volume load are conditions requiring large motor units firing at high rates, which have a strong influence on the sensory and motor cortex modulation of the conscious perception of exertion [37], and which in turn supports the RPE score as an accurate and confident measure of the level of effort when modifying training variables [13,38]. However, this is an assumption partially aligned with the current observation for the HLI protocol, since the current findings supported the negative correlation between RPE scores and repetition number, in spite of the elimination of any possible influence of the volume-load differences between protocols. Thus, there is a variable associated with the level of tensional stimuli when training with high-load intensity that probably influences the high scores of RPE during a complete RT session, which should be further explored in future studies.

In spite of the assumption linking mechanical stress during HLI to the stimulation of fast-twitch fibre recruitment, as is likely to be expected while exercising with a low load in an ischaemic condition [8], the blood lactate concentration in the current study was observed to be higher for LLI+BFR than HLI, suggesting that metabolic stress signalization differed between protocols. Moreover, the [La^−^] responses to both protocols of training (HLI and LLI+BFR) showed values higher than those reported in a previous study comparing unilateral knee extension at 20% 1RM + 100% AOP vs. 80% 1RM (4.2 vs. 4.1 mmol × L^−1^) [15], and were also higher than values reported for unilateral elbow flexion at 20% 1RM + 100 mmHg (5.4 mmol × L^−1^), but similar to the value for elbow flexion at 70% 1RM (7.0 mmol × L^−1^) [23].

Therefore, the higher values of blood lactate accumulation observed for the current LLI+BFR protocol are misaligned with previous findings, suggesting that the glycolytic pathway has a higher demand when exercising with high-load intensities than with low-load intensities [12,14,16]. The current findings are also misaligned with the study of Loenneke et al. [9], in which the blood lactate response for RE protocols performed with and with no BFR showed a tendency to be not different regardless the level of cuff pressure (e.g., 40, 50, or 60% AOP) and the load intensity (i.e., 20, 30, or 70% 1RM). Loenneke et al. [9] also evidenced average lower blood lactate responses (e.g., ~4 vs. ~6 mmol/L) when compared to 20% 1RM (plus different percentages of AOP) with higher pressures and heavier load combinations (e.g., 30% 1RM plus different percentages of AOP). However, these values are lower than those observed in the current study.

Hence, the current findings suggested that training with LLI+BFR significantly increases the [La^−^] values when compared to HLI, probably reinforcing the influence of BFR on lactate metabolism due to the ischemia during exercise [28,39]. But the current results also evidenced that such [La^−^] values during LLI+BFR were significantly higher than HLI training only after the second exercise of the session (i.e., E2), indicating no difference between protocols regarding the [La^−^] response for the first and second exercises (E1). Therefore, only for E1 and E2 did the current results seem to be aligned with previous reports [15,29], simultaneously highlighting the effects of the multiple-exercise performance (i.e., the training session) on the differences in blood lactate accumulation.

Indeed, there are studies reporting that load and the number of repetitions can influence [La^−^]. For example, Rogatzki, Wright, Mikat, and Brice [40] observed that training squat depth with volume load equalised had a tendency to increase blood lactate accumulation (~6 vs. ~4 mmol × L^−1^) when 2 sets/20 repetitions (at 53% 1RM) is compared to 5 sets/5 repetitions (at 85% 1RM) in squat exercise. Balsamo et al. [41] also observed that lactate accumulation is higher (e.g., ~7.5 vs. 6.5 mmol × L^−1^) as the number of repetitions increases following a training session performed with four exercises for large muscle groups (i.e., seated row and bench press) preceding small muscle exercises (e.g., bicep and triceps curl). Moreover, Buitrago et al. [39] reported that blood lactate response increases with the number of repetitions, which is in turn inversely proportional to relative load intensity for a single exercise performance (e.g., bench press exercise: ~5 vs. ~4 mmol × L^−1^ for ~17 repetitions at 55% 1RM vs. ~6 repetitions at 80% 1RM). A similar finding was also observed for a session training planned with five different exercises performed with different loads (e.g., 55 vs. 85% 1 RM), in which blood lactate response (~6 vs. ~5 mmol × L^−1^) was associated with the amount of work performed in a lower-load session (~1.4 vs. ~8.3 tons) [28].

Conversely, there are other studies demonstrating no differences in acute blood lactate accumulation (range from 5.6 to 7.0) when comparing LLI+BFR protocols planned with different loads and volume loads (e.g., 20 vs. 50% 1RM; one vs. three sets; and ~850 to ~2500 kg per session) [42], as well as when comparing large and small muscle groups (femoral quadriceps and brachial biceps) performing four sets at 20% 1RM plus BFR (at 100 mmHg) vs. at 70% 1RM (~9.0 vs. ~7.8 mmol × L^−1^, respectively) [43]. Collectively, these studies favour the notion that the comparisons of the blood lactate responses during LLI+BFR and HLI exercises have shown controversial results, with the differences (or similarities) varying according to the planning of training variables (i.e., volume load, load, number of repetitions, cuff pressure, exercise duration, and mode of execution). From the current findings, a training session (i.e., multiple exercises planning) with low loads was also a factor determining blood lactate responses between different protocols, despite no significant correlations being observed between blood lactate and variables of training.

A last important concern to be considered is whether rest intervals have probably influenced lactate production and/or accumulation when comparing HLI and LLI+BFR protocols. However, blood lactate accumulation did not differ (i.e., ~7.0 vs. ~7.3 vs. ~6.9 mmol/L) between training sessions planned with six bilateral multi-joint exercises for upper- and lower-limbs, rest intervals of 30, 60, and 90 s, similar volume (four sets, ten repetitions to maximum), but a different volume load (~26 vs. ~29 vs. ~30 tons) [44]. Moreover, blood lactate accumulation also seemed not to differ (i.e., ~8.8 vs. ~8.0 vs. ~8.7 mmol × L^−1^) in response to training sessions planned with different load intensities (60, 70, and 90% 1RM), a similar number of exercises (eight exercises involving upper and lower limbs), a similar total number of repetitions (i.e., 15 rep × 2 sets, 3 rep × 3 sets, and 5 rep × 6 sets), and rest intervals (120 s), in spite of a different volume load (~9.8 vs. ~15.0 vs. ~17.5 tons) [45]. Hence, the observed differences in blood lactate accumulation between current protocols cannot be attributed to the planning of training variables, mainly because both protocols (HLI and LLI+BFR) follow the recommendations for optimal stimuli on muscle mass and strength [6,20]. In addition, the lack of significant correlation between the variables of training and blood lactate responses to both protocols (except for repetition number) further supported the notion that rest period did not favour blood lactate response.

A final assumption to be addressed is the suggested mechanism underlying higher blood lactate accumulation, which is probably related to the different intramuscular dynamics of lactic acid when training with HLI and LLI exercise. According to Buitrago et al. [39], HLI exercise activates glycolysis at high rates, with no high accumulation of lactate due to the cell-to-cell shuttle (i.e., type II fibres release and type I fibres reuptake and oxidation), whereas the low rate of lactate production during LLI (without BFR) might cause accumulation over time due to the longer time of the exercise. Despite the current study not analysing lactic acid production and release, this might be the mechanism at play in the current study when analysing blood lactate concentration, since LLI+BFR protocol performance lasted 24% more than HLI. Curiously, the LLI+BFR protocol duration was poorly and non-significantly related to [La^−^] response, but the [La^−^] responses are highly correlated between protocols, suggesting that the HLI protocol stimulated anaerobic metabolism through a similar mechanism, but at a lower level than the LLI+BFR protocol.

While the current results cannot verify whether duration [39] or lactate balance [46] (or both) determined the metabolic differences between HLI and LLI+BFR protocols, a recent study evidenced that metabolism during RT sessions (measured by the amount of O_2_ consumed) is higher in low-load protocols than for high-load protocols, which was a result credited to the exercise duration instead of volume or repetition differences [47]. Hence, for practical purposes, the current findings suggested that training with low-load intensity and BFR might be an alternative to traditional high-load resistance training when aiming to exercise demanding a high level of exertion, which is a condition previously associated with improvements in muscle strength and mass [8], since higher metabolic stimulation is considered a reasonable signal for fibre type II recruitment, as well as anabolic metabolism activation [1,4].

The current study has the following limitations:(i)The lack of a third protocol planned with low-load training without BFR to compare the effect of RPE and [La^−^] responses with the LLI+BFR, which would probably support the question of whether BFR is the only condition eliciting high blood lactate accumulation when training with low load intensity.(ii)The trained young adults usually show a high tolerance threshold for pain and discomfort during RE, which suggests that future studies should explore if sex, age, and strength conditioning level differences can affect RPE and [La^−^] responses similarly.(iii)The lack of a subgroup of participants experienced with low-load resistance training could mean the unusual (or novelty) exercise condition is not a confounding factor when comparing the RPE with a high-load intensity training or exercise condition.(iv)The use of SBP instead of AOP to define cuff pressure during training might be a concern since the effects of the BFR can be underestimated with actual pressure. However, the current mean cuff pressure during training (60 ± 2 mmHg) is within the range of the absolute cuff pressures at 40 to 50% of AOP (55 ± 7 mmHg and 61 ± 5 mmHg, respectively) used in the studies of Lixandrão et al. [21] and Moriggi et al. [48]. Moreover, given the theoretical range for OAP (100–210 mmHg) [49], the 50% SBP might probably rely on the zone of cuff pressures (40–80% AOP) in which BFR effects can be considered ideal to optimise BFR effects and reduce risks [50], despite the differences between these two variables regarding values and procedures [51].(v)The volume load difference between protocols is a concern, since the higher volume load may influence the high RPE scores during HLI protocols. Despite the findings of previous studies regarding the influence of volume load on RPE and blood lactate responses being conflicting, the HLI training was perceived as heavier in being performed than LLI+BFR in the current study, with the RPE response inversely associated with metabolic response and the number of repetitions (i.e., a variable of volume).

However, the way that the magnitude of volume load might be the factor influencing RPE scores should be further explored in future studies, since it is relevant that the observation of the procedures to quantify volume load during resistance training with BFR have never considered the effect of cuff pressure as a variable of exercise intensity. Thus, cuff pressure should be an issue to be considered for a better equalisation of volume load between resistance training with and without BFR in future studies, since studies have repeatedly demonstrated the influence of pressure level on exercise intensity [7,10].

## 5. Conclusions

The present study demonstrated that RT session planned with high- and low (+BFR)- load intensities and multiple exercises modified RPE and blood lactate responses, with both values reaching levels corresponding to high or very-high exercise intensity in the actual participants (i.e., young male adults). Specifically, the findings evidenced that training with a high load increases RPE scores till (or close to) maximal ratings, which was observed to be higher as lower repetitions were performed, but not related to the volume load of the training. This association suggested that a training session planned with a high load and demanding lower repetitions is perceived as a heavier exercise condition, which was observed to be independent of metabolic stress (e.g., level of blood lactate accumulation). In contrast, training with a low load and BFR was perceived as less heavy to be performed (i.e., a hard but not very-hard exercise condition when compared to the RPE scores for HLI training), in spite of the higher blood lactate accumulation. This finding suggests that the level of metabolism activation during resistance training is not indicative of how heavy training is perceived, despite the blood lactate concentration attained values corresponding to a very-high intensity muscular effort (i.e., >8 mmol × L^−1^).

Therefore, both protocols (HLI and LLI-BFR) of training produced distinguishable results from previous findings concerning RPE and [La^−^] responses, despite both current training protocols evidencing typical short high-intensity training exertion from perceptual and metabolic perspectives, confirming the hypothesis of the current study. Thus, the current findings support the recommendation of low-load plus BRF to provide high-intensity resistance training (from a metabolic perspective) independently of an individual’s physical abilities or disabilities. This finding is clinically relevant for individuals with an inclination to avoid heavy weights since the current responses from a complete resistance training section indicated that LLI+BFR is effective at increasing muscle metabolism, despite the chronic effect on muscle mass and strength still remaining unaddressed, as well as the extrapolation to populations other than the current sample of young male adults.

Thus, the inferences of the findings involve particularities of young men, therefore requiring additional studies for a generalised population (i.e., different age and sex groups, as well as training experience level). Future studies should even assess the long-term effect of LLI+BFR on muscle hypertrophy, strength, and injury risk, particularly in clinical populations and ageing adults, but not only since the effect on athletic performance (e.g., tolerance, power, economy, and recovery) remain unaddressed. In addition, wearable technology could be leveraged to monitor real-time physiological responses and adaptations to training.

## Figures and Tables

**Figure 1 sports-13-00148-f001:**
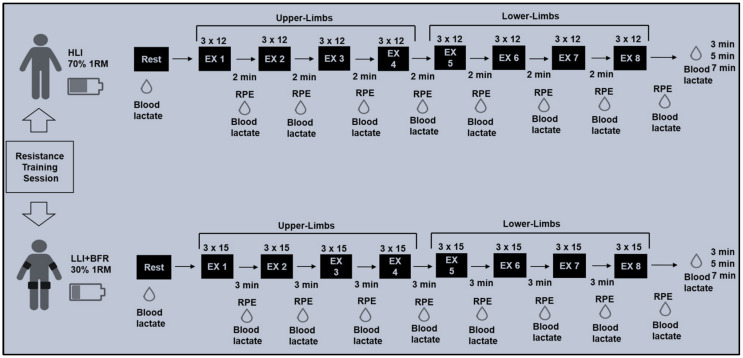
Training protocol overview. Obs.: The exercise numbers in the black boxes correspond to the following exercises: Ex1 (flat bench press), Ex2 (seated row), Ex3 (triceps curl), Ex4 (biceps curl), Ex5 (single leg extension), Ex6 (prone leg curl machine), Ex7 (leg press 45°), and Ex8 (rack calf raises). The RPE and [La^−^] acronyms refer to the rate of perceived exertions and blood lactate samples, respectively), and SBP refers to systolic blood pressure.

**Figure 2 sports-13-00148-f002:**
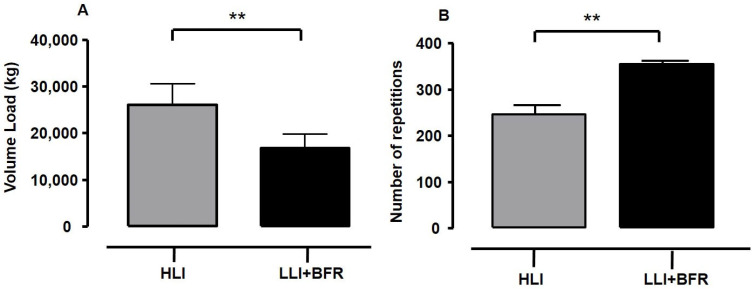
The volume load (**Panel A**, in kilograms—kg) and total number of repetitions (**Panel B**) in each training protocol. High-intensity load training protocol (grey bars: HLI, N = 14, 95% CI 23,475.3–28,669.9 kg, and 234.4–257.3 repetitions); low-intensity load with blood flow restriction training protocol (dark bars: LLI+BFR, N = 14, 95% CI 15,098.1–18,573.6 kg, and 350.9–359.3 repetitions). ** *t*-test significant difference at *p* ≤ 0.01 between training protocols.

**Figure 3 sports-13-00148-f003:**
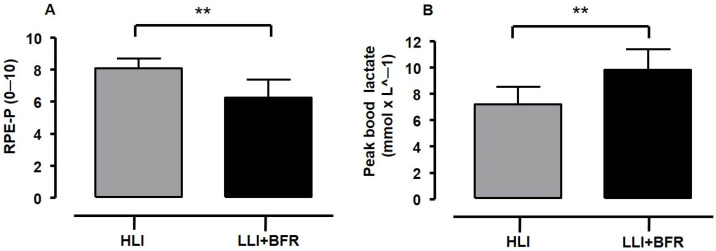
The average RPE (0–10 scale) for each training protocol (**Panel A**) and blood lactate peak after each training protocol (**Panel B**). High-intensity load (grey bars: HLI, N = 14, 95% CI 7.7–8.4 rating, and 6.4–7.9 mmol × L^−1^), low-intensity load with blood flow restriction (dark bars: LLI+BFR, N = 14, 95% CI 5.5–6.8 rating, and 8.9–10.7 mmol × L^−1^). ** *t*-test significant difference at *p* ≤ 0.01 between protocols.

**Figure 4 sports-13-00148-f004:**
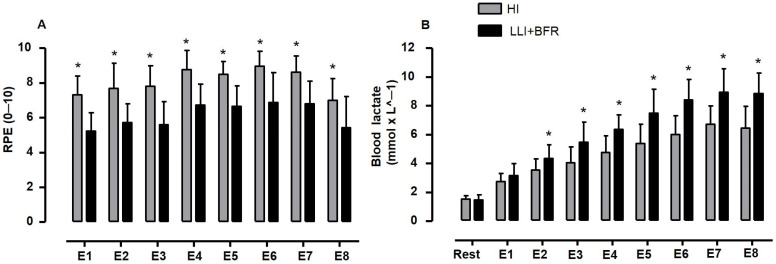
Comparisons of RPE (0–10 scale) and blood lactate concentration ([La^−^] in mmol × L^−1^) (**Panels A** and **B**, respectively) between each pair of exercises in HLI (grey bars) and LLI+BFR (black bars) protocols. * Two-way repeated-measure ANOVA significant difference at *p* ≤ 0.05 between exercises, where the following abbreviations are used: Ex1 (flat bench press), Ex2 (seated row), Ex3 (triceps curl), Ex4 (biceps curl), Ex5 (single leg extension), Ex6 (prone leg curl machine), Ex7 (leg press 45°), and Ex8 (rack calf raises).

**Table 1 sports-13-00148-t001:** Resistance training protocols.

Variables of Planning	Protocols	
	HLI	LLI+BFR
Load	70% 1RM	30% 1RM
Sets	3	3
Repetitions	12	15
Rest (between sets in seconds)	60	30
Rest (between exercises in seconds)	120	180

Obs.: HLI (high-load intensity), LLI+BFR (low-load intensity plus blood flow restriction), and 1RM (one repetition to maximum).

**Table 2 sports-13-00148-t002:** Rating of perceived exertion (RPE) and blood lactate ([La^−^]) responses throughout the training protocols.

		RPE (0–10 Scale)		[La^−^] (mmol × L^−1^)	
		HLI	LLI+BFR	HLI	LLI+BFR
	**Rest**	-	-	1.5 ± 0.2 ^a,b,c,d,e,f,g,h^	1.4 ± 0.3 ^a,b,c,d,e,f,g,h^
**Upper-limbs**	**E1**	7.3 ± 1.1 ^d,e,f,g^	5.3 ± 1.1 ^d,e,f,g^	2.7 ± 0.6 ^b,c,d,e,f,g,h^	3.1 ± 0.8 ^b,c,d,e,f,g,h^
	**E2**	7.7 ± 1.4	5.8 ± 1.2	3.5 ± 0.8 ^d,e,f,g,h^	4.3 ± 1.0 ^c,d,e,f,g,h^
	**E3**	7.8 ± 1.2	5.6 ± 1.5 ^d,e,f^	4.0 ± 1.1 ^d,e,f,g,h^	5.5 ± 1.4 ^d,e,f,g,h^
	**E4**	8.8 ± 1.1 ^h^	6.7 ± 1.4	4.7 ± 1.2 ^g,h^	6.3 ± 1.0 ^e,f,g,h^
**Lower-limbs**	**E5**	8.5 ± 0.7 ^h^	6.6 ± 1.2	5.4 ± 1.3 ^g,h^	7.5 ± 1.7 ^f,g,h^
	**E6**	8.9 ± 0.9 ^h^	6.9 ± 1.8 ^h^	6.0 ± 1.2	8.4 ± 1.4
	**E7**	8.6 ± 0.9 ^h^	6.9 ± 1.3 ^h^	6.7 ± 1.3	8.9 ± 1.6
	**E8**	7.0 ± 1.3	5.4 ± 1.8	6.4 ± 1.5	8.8 ± 1.4

Obs.: HLI = high-load intensity training protocol; LLI+BFR = low-load intensity with blood flow restriction training protocol; (E1) flat bench press; (E2) seated row; (E3) triceps curl; (E4) biceps curl; (E5) single leg extension; (E6) prone leg curl machine; (E7) leg press 45°; and (E8) rack calf raises. Two-way repeated-measure ANOVA significant difference between exercises at *p* ≤ 0.05: ^a^ different from E1, *p* < 0.05; ^b^ different from E2, *p* < 0.05; ^c^ different from E3, *p* < 0.05; ^d^ different from E4, *p* < 0.05; ^e^ different from E5, *p* < 0.05; ^f^ different from E6, *p* < 0.05; ^g^ different from E7, *p* < 0.05; ^h^ different from E8, *p* < 0.05.

## Data Availability

The data that support the findings of this study are available from the last author (dalton.pessoa-filho@unesp.br) upon reasonable request.

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
