# Peer review of "Perceptual and Metabolic Responses During Resistance Training Sessions: Comparing Low-Load Plus Blood Flow Restriction with High-Load Plans"

_sports, 2025, doi:10.3390/sports13050148_

Round 1
Reviewer 1 Report
Comments and Suggestions for Authors
Dear authors: Congratulations on your work. I think it is interesting to advance the relationship between perceived exertion and blood lactate concentration to qualify the level of exertion during endurance exercise, in this case, with vascular flow restriction.
In my view, the introduction adequately reviews the problem statement and clearly defines the objective of the study. The description of the participants is clear while the Strength Measurements and Training Protocols are well defined.
I have a discrepancy in line 199, (and line 423) as I understand that all participants were male and their age was 26 years (+/- 2.6). I understand that the age and sex observations are superfluous.
I think the protocols are well specified in the two experimental conditions. Figure 1 clearly illustrates the chronology of lactate recording. The Metabolic and perceptual measurements are well written and are in line with the biliography for this type of measurement.
The statistical analysis (Two-way analysis of variance ANOVA) is appropriate for this type of study.
The presentation of the results, including figures 2, 3 and table 1, give a good summary of the results.
The discussion is well written in relation to the results obtained and the conclusions are also consistent with the results.
I would only recommend to the authors that, in order to improve the manuscript, you could include a paragraph of limitations about their study, as well as future implications.
Sincerely yours.
Author Response
Dear Reviewer 1
My fellow authors and I would like to thank you for revising this manuscript and for the comments, which has enriched it substantively. We believe we have adequately addressed each of the comments. However, if you deem more changes necessary, we look forward to addressing any other concerns.

Reviewer 2 Report
Comments and Suggestions for Authors
- Title & Abstract
Issues:
The title is clear but long—consider shortening it for better impact.
The abstract is informative but contains complex sentences that reduce readability.
Results need more contextualization to highlight their practical implications.
Suggested Revisions:
Shorten the title while maintaining clarity. Example:
"Perceptual and Metabolic Responses to Low-Load Blood Flow Restriction vs. High-Load Training"
Simplify complex sentences and ensure key findings are highlighted.
Clarify the significance of findings for training applications.
Example Revision for Abstract Conclusion:
Instead of:
"The RPE and [La-] responses to HLI and LLI+BFR suggested that protocols are distinguishable regarding perceptual and metabolic stimuli."
Try:
"Our findings suggest that while LLI+BFR is metabolically demanding, it is perceived as easier than HLI, indicating its potential as an alternative resistance training strategy, especially for populations unable to tolerate high loads."
- Introduction
Issues:
The research gap needs to be more explicitly stated before the objectives.
The literature review is informative but could be more concise.
The hypothesis is stated late—introduce it earlier for better flow.
Suggested Revisions:
State the research gap clearly:
Example:
Instead of:
"Although individual studies have examined RPE and lactate responses to different training loads, a direct comparison between full training sessions with HLI vs. LLI+BFR remains unexplored."
Try:
"While RPE and lactate responses to isolated exercises have been studied, their cumulative effects over a full training session with HLI versus LLI+BFR remain unclear. This study aims to fill that gap."
Introduce the hypothesis earlier for better clarity.
- Methods
Issues:
The participant selection criteria lack details to clarify eligibility.
Statistical methods should be justified more explicitly.
The intervention description is comprehensive but could be structured more clearly.
Suggested Revisions:
Clarify participant eligibility criteria:
Example:
Instead of:
"Fourteen trained adults participated."
Try:
"Fourteen recreationally trained adults (26.2 ± 2.6 years, training ≥3x per week) were recruited. Exclusion criteria included cardiovascular conditions, metabolic disorders, and a BMI ≥30 kg/m²."
Summarize intervention protocols in a table for clarity.
Justify statistical methods explicitly:
Example:
Instead of:
"Two-way ANOVA was used to compare RPE and [La-] responses."
Try:
"A two-way ANOVA was applied to assess differences between exercise conditions while accounting for repeated measures, ensuring appropriate statistical control."
- Results
Issues:
The tables are data-heavy and could be complemented with more visual representations.
Statistical results should be better interpreted—explain what the numbers mean in practical terms.
Effect sizes should be emphasized to provide context for significance.
Suggested Revisions:
Add a summary before tables to highlight key trends.
Use visual aids (e.g., bar charts) to complement numerical data.
Interpret effect sizes for clarity:
Example:
Instead of:
"Condition C4 reduced rearfoot peak force (-139.09 N, p < 0.001)."
Try:
"The combined intervention (C4) significantly reduced rearfoot peak force (-139.09 N, p < 0.001), with a large effect size (Cohen’s d = 1.02), indicating a meaningful reduction in impact forces."
- Discussion
Issues:
Too many citations are clustered together, making it harder to follow the argument.
Comparison with previous studies could be expanded for better scientific grounding.
Practical applications must be more precise for coaches, athletes, and clinicians.
Suggested Revisions:
Reduce excessive in-text citations and integrate them more smoothly.
Expand on comparisons with previous research to highlight differences.
Provide clear, actionable recommendations for training.
Example Revision for Practical Applications:
Instead of:
"These findings suggest that combining cadence adjustments, footwear modifications, and orthoses could enhance injury prevention and running efficiency."
Try:
" LLI+BFR may provide a viable alternative to traditional resistance training by eliciting similar metabolic demands with lower mechanical stress for individuals unable to tolerate high loads. This approach could be particularly beneficial for rehabilitation programs and older adults."
- Conclusion
Issues:
The conclusion effectively summarizes key findings but lacks a strong take-home message.
Future research directions should be more explicitly stated.
Suggested Revisions:
Emphasize practical applications more strongly.
Clearly outline future research directions.
Example of a Stronger Conclusion:
Instead of:
"Future studies should explore long-term adaptations to these interventions in real-world running environments."
Try:
"Future research should assess the long-term impact of LLI+BFR on muscle hypertrophy, strength, and injury risk, particularly in clinical populations and aging adults. Additionally, wearable technology could be leveraged to monitor real-time physiological adaptations."
- References
Issues:
Ensure consistency in reference formatting (e.g., journal abbreviations, DOI links).
Some older references should be updated to reflect recent findings.
Suggested Revisions:
Check formatting for consistency and completeness.
Replace outdated references with more recent studies where applicable.
Final Recommendations
Key Areas to Improve:
Improve readability by simplifying complex sentences.
Clarify the research gap and highlight the study’s significance early.
Summarize training protocols in a table for better comprehension.
Enhance statistical interpretation and discuss effect sizes.
Use more visual aids (e.g., figures and bar charts) to complement dense data tables.
Expand the discussion on practical applications for broader audience impact.
Author Response
Dear Reviewer 2
My fellow authors and I would like to thank you for revising this manuscript and for the comments, which has enriched it substantively. We believe we have adequately addressed each of the comments. However, if you deem more changes necessary, we look forward to addressing any other concerns.

Reviewer 3 Report
Comments and Suggestions for Authors
Thank you for the opportunity to review this manuscript for your journal. This article addresses a highly relevant topic in resistance training with potential applications in health and aging contexts. Additionally, the literature review provided is extensive and informative. However, it is somewhat disorganized and lacks a clear connection to the study's proposed objectives.
Major concerns:
1. The introduction appears disorganized, and both the research question and study hypothesis remain unclear.
The focus shifts too early to Rating of Perceived Exertion (RPE) and blood lactate concentration. Instead, it may be more relevant to first discuss the general benefits of resistance training, particularly the differences between high-load and low-load training. Following this, blood flow restriction training (BFR) with low loads should be introduced as an innovative strategy to simulate the metabolic stress induced by high loads. Prior studies applying this technique in populations unable to lift heavy loads should be highlighted. Lastly, the introduction should emphasize the importance of load quantification in resistance training and justify the use of blood lactate concentration and RPE as key assessment methods.
2. The implementation of the LLI+BFR protocol should be based on previous studies, and the rationale for adjusting to arterial occlusion pressure during exercise and rest must be better detailed and justified.
3. Performing 3 sets of 12 repetitions at 70% of 1RM often leads to near muscular failure. Were all participants able to complete all repetitions across all sets? In the discussion (line 276), it is suggested that participants in the high-load condition trained to failure. This needs to be clarified in the methodology section.
4. For the LLI+BFR protocol, were repetitions in reserve (RIR) recorded? RPE after each set is more closely related to RIR than to absolute load percentage. Was this aspect considered in the study design? How many RIR were left in the LLI+BFR? And in the studies cited?
5. Please remove all references to trends in the results section, as well as any interpretation of the findings. Results should be presented objectively, with interpretations left for the discussion.
6. The discussion is overly long and lacks organization. Since the study hypothesis was not clearly stated at the beginning, the discussion presents numerous previous studies without a clear argumentative thread relating to the main research question. Please refine the research question, improve the justification of methodological choices, and discuss only the aspects investigated in this study. The discussion should begin with the study’s key findings and compare them to existing evidence. Any argumentation related to aspects not investigated in the present study should be excluded from the discussion section for the sake of clarity.
7. The first sentence of the conclusions appears out of context. Was the objective of this study to compare a multiple-exercise plan with a single-exercise plan? If not, this should be revised.
Minor comments:
Line 69: Please correct "LLI+BFR".
Line 98: Please correct "males".
Line 133: Please revise "BFR procedures was".
Overall, the study covers an important topic, but significant revisions are needed to clarify its objectives, improve methodological details, and ensure coherence in the discussion.
Comments on the Quality of English LanguageShould be improved
Author Response
Dear Reviewer 3
My fellow authors and I would like to thank you for revising this manuscript and for the comments, which has enriched it substantively. We believe we have adequately addressed each of the comments. However, if you deem more changes necessary, we look forward to addressing any other concerns.

Round 2
Reviewer 2 Report
Comments and Suggestions for Authors
- Basic Reporting
Strengths:
- The article follows the IMRaD structure (Introduction, Methods, Results, and Discussion).
- Key components such as ethical approval and conflict of interest are present.
🔧 Revisions Needed:
- Language and Grammar: The manuscript would benefit from language editing for grammar, tense consistency, and flow. Some sentences are awkward or overly long.
- Abbreviations: All abbreviations (e.g., HRV, RPE, VO₂max) must be defined upon first use in the abstract and again in the main text if needed.
- Figures and Tables:
- Ensure all tables and figures are referenced in-text in the correct order.
- Improve the clarity of figure captions by including units, group identifiers, and statistical annotations.
- Double-check that all visual elements are high-resolution and legible.
- References:
- Could you make sure all citations are formatted consistently per journal style?
- Could you check that all references cited in the text are included in the reference list?
- Experimental Design
Strengths:
- The crossover design with a washout period is appropriate for the study goals.
- Participant eligibility and exclusion criteria are clearly stated.
Revisions Needed:
- Sample Size Justification: Include a power analysis with assumptions (effect size, alpha, power) to justify the sample size.
- Randomization: Describe the method used for random allocation to conditions and how carryover effects were minimized.
- Blinding: Clarify whether participants, trainers, or outcome assessors were blinded to the intervention order.
- Intervention Description:
- Could you include a detailed protocol for the intervention and control sessions (e.g., warm-up duration, exercise intensity)?
- Could you clarify whether exercise intensity was matched between sessions?
- State how RPE and HR were measured and at what time points.
Suggestion:
“A priori power analysis was performed using G*Power (version X) to estimate the sample size needed to detect a moderate effect (Cohen’s d = 0.5) with 80% power and α = 0.05, resulting in a required n = 14.”
- Validity of the Findings
Strengths:
- Statistical tests appear appropriate for within-subject comparisons.
- Significant outcomes are reported clearly.
Revisions Needed:
- Statistical Reporting:
- Provide exact p-values and corresponding effect sizes (e.g., Cohen’s d, η²).
- Report confidence intervals for key findings.
- Clearly describe the test used for each outcome (e.g., paired t-test, Wilcoxon).
- Interpretation of Results:
- Could you discuss whether the observed changes are clinically meaningful, not just statistically significant?
- Could you acknowledge the potential for Type I error due to multiple comparisons (consider Bonferroni correction or similar)?
- Limitations:
- Discuss the short-term nature of the study, sample size limitations, and lack of generalizability.
- If only male participants were used, acknowledge sex-based limitations.
Suggestion:
Although statistically significant, the differences in HRV post-session may not exceed thresholds for clinical relevance. Future research should explore chronic adaptations across a training cycle.”
Author Response
Dear Reviewer
Please, see the responses to your comments and questions in the file attached.
Thank you very much for your revision.

Reviewer 3 Report
Comments and Suggestions for Authors
I would like to thank the authors for their thoughtful and thorough responses to the comments and suggestions provided in the previous review. The revisions made have substantially improved the clarity and quality of the manuscript.
All of my concerns have been adequately addressed, and I now find the manuscript suitable for publication. I support its progression to the next stage of the editorial process.
Please check "RPR" in line 293
Author Response
My fellow authors and I would like to thank you for the new set of comments, which have enriched the manuscript even more. Once again, we believe we have adequately addressed each of the comments. However, if you deem more changes necessary, we look forward to addressing any other new concern.
We checked the text to revise all RPE acronyms.
We also rearragend the results (please see lines 245 to 303), and inserted additional information in the conclusion (please, see lines 527 to 560) in an attempt to improve both sections.
Thank you very much again for your revision.